# The Incidence of Associated Anomalies in Children with Congenital Duodenal Obstruction—A Retrospective Cohort Study of 112 Patients

**DOI:** 10.3390/children9121814

**Published:** 2022-11-24

**Authors:** Adinda G. H. Pijpers, Laurens D. Eeftinck Schattenkerk, Bart Straver, Petra J. G. Zwijnenburg, Chantal J. M. Broers, Ernest L. W. Van Heurn, Ramon R. Gorter, Joep P. M. Derikx

**Affiliations:** 1Department of Pediatric Surgery, Emma Children’s Hospital Amsterdam UMC, University of Amsterdam, Meibergdreef 9, 1105 AZ Amsterdam, The Netherlands; 2Department of Pediatric Cardiology, Emma Children’s Hospital Amsterdam UMC, University of Amsterdam, Meibergdreef 9, 1105 AZ Amsterdam, The Netherlands; 3Department of Clinical Genetics, Emma Children’s Hospital Amsterdam UMC, University of Amsterdam, Meibergdreef 9, 1105 AZ Amsterdam, The Netherlands; 4Department of Pediatrics, Emma Children’s Hospital Amsterdam UMC, University of Amsterdam, Meibergdreef 9, 1105 AZ Amsterdam, The Netherlands

**Keywords:** duodenal obstruction, duodenal atresia, cardiac anomalies, associated anomalies, VACTERL

## Abstract

Background: Duodenal obstruction (DO) is a congenital anomaly that is highly associated with other anomalies, such as cardiac anomalies and trisomy 21. However, an overview of additional anomalies and patient-specific risk factors for cardiac anomalies is lacking. Potential association with the vertebral, anorectal, cardiac, trachea-esophageal, renal and limb anomalies (VACTERL) spectrum remains unknown. Therefore, we aim to examine the incidence of associated anomalies, a VACTERL-spectrum association and patient-specific risk factors for cardiac anomalies in patients with DO. Methods: A retrospective cohort study was performed between 1996 and 2021. Outcomes were the presence of any additional anomalies. Risk factors for cardiac anomalies were analyzed using multivariate logistic regression. Results: Of 112 neonates with DO, 47% (N = 53/112) had one associated anomaly and 38% (N = 20/53) had multiple anomalies. Cardiac anomalies (N = 35/112) and trisomy 21 (N = 35/112) were present in 31%. In four patients, VACTERL-spectrum was discovered, all with cardiac anomalies. Trisomy 21 was found to be a risk factor for cardiac anomalies (OR:6.5; CI-95%2.6–16.1). Conclusion: Associated anomalies were present in half of patients with DO, of which cardiac anomalies and trisomy 21 occurred most often, and the VACTERL-spectrum was present in four patients. Trisomy 21 was a significant risk factor for cardiac anomalies. Therefore, we recommend a preoperative echocardiogram in patients with DO. In case a cardiac anomaly is found without trisomy 21, VACTERL-screening should be performed.

## 1. Introduction

Duodenal obstruction (DO) is a congenital anomaly caused by duodenal atresia (DA), web or membrane or annular pancreas (AP), and occurs in 1 in 5000–10,000 live births [1,2]. DO most commonly presents as an isolated anomaly; however, it has also been associated with various other congenital anomalies. Cardiac anomalies and trisomy 21 are the most frequently documented, which both account for almost one-third of patients with DO according to previous studies [3,4]. The incidence of cardiac anomalies without the presence of trisomy 21 remains unknown, as well as the potential risk factors for cardiac anomalies. Information about the potential associated cardiac anomalies prior to surgery is essential in patients with DO, as they may have serious (anesthesiological) consequences. It is important to have full knowledge of the associated cardiac anomalies, since severe cardiac anomalies can have a major influence on the outcome and care of patients with DO [5].

Although there is a strong correlation between DO, trisomy 21 and cardiac anomalies, there is a lack of knowledge of other potential associated anomalies. Previous studies suggest a potential association with esophageal atresia, anorectal malformations or vertebral anomalies and DO [6,7,8,9]. These anomalies are all part of the (vertebral anomalies, anal atresia, cardiac anomalies, tracheo-esophageal fistula, renal anomalies, limb abnormalities (VACTERL) spectrum [10]. It might be that DO patients with additional congenital anomalies are actually part of the VACTERL spectrum, which at the moment is unknown. Moreover, an overview of all anomalies that might be associated with DO is lacking. Knowledge about potential associated anomalies and VACTERL spectrum is crucial in order to determine whether or not screening for them is necessary. When screening is not a part of routine care, this could potentially delay the diagnosis and treatment of additional anomalies of the VACTERL spectrum. However, performing routine screening in all patients seems not cost-effective. It is therefore important to look for patient specific risk factors which increase the chances of finding additional anomalies.

Therefore, the aim of this study is to: (1) provide a comprehensive overview of all associated anomalies present in our cohort of patients with DO, (2) see if there is a VACTERL spectrum association, and (3) determine patient-specific risk factors for these additional anomalies. We hypothesize that cardiac anomalies and trisomy 21 are the most frequent associated anomalies in patients with DO.

## 2. Materials and Methods

All patients with a DO, defined as duodenal atresia, duodenal web or annular pancreas, treated between July 1996 and December 2021, were retrieved from the Amsterdam University Medical Centers (AUMC) database. The AUMC consists of two pediatric surgical centers, namely, Academical Medical Center (AMC) and VU University Medical Center (VUmc), and is a center of expertise for rare and complex gastroenterological diseases. Patients with a malrotation as the cause of DO without atresia, web/membrane or annular pancreas, or patients with pyloric atresia in the surgical report, were excluded from the study. The local medical ethical commission evaluated and validated the study (W18_233#18.278, 26 July 2018). An opt-out letter was sent to patients and parents, which they could return within one month if they did not wish to participate. Following this period, patient records were checked.

Two PhD candidates (LES and AP) read the patient records and double-checked 10% of each other’s work. In case of doubt, a paediatric surgeon (JD) was consulted. All data for patients meeting the inclusion criteria were restored in SPSS 28.0.

The primary outcome for this study was the presence of any associated anomaly next to DO. Secondary outcomes concerned the patient-specific risk factors for cardiac anomalies and the presence of VACTERL-spectrum.

Not all children with DO received a routine echocardiogram in our cohort. All patients with trisomy 21 received an echocardiogram, as well as symptomatic patients. From 2012, all children with DO received an echocardiogram. Further research was conducted investigating the VACTERL spectrum when the echocardiogram was abnormal. Based on this indication, further additional diagnostics were performed. A clinical geneticist was consulted in case of numerous congenital anomalies.

### 2.1. Data Extraction

The patients were categorized by the type of DO namely atresia, annular pancreas or web/membrane. This and the type of surgical technique were extracted from the surgical reports. Furthermore, patient information concerning sex, gestational age in weeks, prematurity (defined as gestational age < 37 weeks), age at surgery in days, birthweight, trisomy 21, length of hospital stay in days, length of follow-up in months and mortality within 30 days after surgery were extracted.

Information concerning associated anomalies was divided in organ systems: cardiovascular, vertebral, renal, genito-urinary tract, eye ear and neck, esophageal, gastro-intestinal, limbs, cleft, lip and palate, nervous system and genetics. Specific details were reported. Patients meeting the criteria of the VACTERL association were defined following the definitions of Van de Putte et al. [10]. The congenital anomalies are divided into major and minor anomalies. Major VACTERL features are part of the classical VACTERL association. Minor VACTERL features are anomalies that are not classically seen in VACTERL association, but appear in the same organ systems as typical VACTERL features.

Based on the major and minor criteria, four subtypes of VACTERL were created: STRICT-VACTERL, VACTERL-LIKE, VACTERL-PLUS and NO-VACTERL.

STRICT-VACTERL contains three or more major VACTERL anomalies, no anomalies outside VACTERL. VACTERL-LIKE contains three or more major and minor VACTERL anomalies, with no anomalies outside the VACTERL spectrum. VACTERL-PLUS contains all cases that fulfil the criteria of STRICT-VACTERL and VACTERL-LIKE and also have anomalies outside of the VACTERL spectrum. Cases that did not meet the VACTERL subtypes mentioned above are included in the NO-VACTERL subtype. For further analysis, a dichotomous variable was formed, defining VACTERL-STRICT, VACTERL-LIKE and VACTERL-PLUS as presence of VACTERL, and NO-VACTERL as not present. In case of doubt, a clinical geneticist was consulted (PZ).

Cardiac anomalies were reported by presence and type following the definitions of Putte et al., and Cunningham et al. [10,11]. Isolated patent ductus arteriosus and patent foramen ovale not requiring surgery under gestational age of 37 weeks were not registered as cardiac anomalies. Two patients had an asymptomatic double vena cava without cardiological follow-up and were not registered as cardiac anomalies. In case of doubt, a pediatric cardiologist (BS) was consulted.

### 2.2. Statistical Analysis

Normally distributed variables were reported with mean and ± standard deviation (SD) and non-normally distributed variables were reported as median with interquartile range. Distribution was tested by the comparison of a histogram of the sample data with a normal probability curve. To test the significance of the differences in categorical data the Chi-squared test was used; if the assumptions of this test were not met, then the Fisher’s Exact test was used. In case of continuous data, Student’s *t* test was used. Statistical significance was defined as *p*-values of ≤ 0.05.

Multivariate logistic regression analysis was used for the outcome of associated cardiac anomalies, including trisomy 21, prematurity and multiple birth as input variables. Backward Wald selection was used for the selection of variables using the standard *p* = 0.10 for variable removal. Significant risk factors were reported using odds ratio (OR) with 95% confidence intervals (95%-CI). An assessment of confounding (increase in B-coefficient of >10%) and effect modification (significant interaction term) was performed. Additionally, the adjusted R-squared was reported to show the proportion of the variance in the occurrence of cardiac anomalies explained by the model.

## 3. Results

A total of 112 patients with DO were operated in Amsterdam in the study period and met the inclusion criteria. Seven patients were excluded due to missing data regarding associated anomalies. No opt-out letters were returned within one month. Baseline characteristics are shown in Table 1.

### 3.1. Patient Characteristics

Among all patients, 37% (N = 41/112) were treated for duodenal atresia, 33% (N = 37/112) for duodenal web or membrane and 30% (N = 34/112) for annular pancreas. Sex was nearly equally distributed, with 52% males (N = 58/112). Prematurity was seen in 48% of patients (N = 51/106, missing: 6). Data for birthweight were missing in seven patients, giving a median weight of 2620 g (IQR: 2080–3090).

The majority of the DO patients were treated with a duodeno-duodenostomy (N = 88/112); the patients with web or membrane (62% N = 23/37) were treated by duodenoplasty and one patient was treated with a duodeno-jejunostomy. In 2% (N = 2/112) of the patients, a transanastomotic tube was placed during surgery. All other patients received a nasogastric tube perioperative. The median age at surgery was 3 days (IQR: 1–6). The median hospital stay was 23 days (IQR: 18–33) and the median time of follow-up was 33 months (IQR: 6–108).

Mortality within 30 days was seen in 3% (N = 3/112) of the patients. All three were diagnosed with trisomy 21, of which two diagnoses were combined with severe cardiac anomalies such as atrioventricular septal defect (AVSD). One patient with only trisomy 21 died as a result of ongoing sepsis due to an anastomotic leakage after duodeno-duodenostomy.

### 3.2. DO and Cardiac Anomalies

Cardiac anomalies were seen in 31% (N = 35/112) of the patients. Twenty out of these 35 patients (57%) were also diagnosed with Trisomy 21. The incidence of cardiac anomalies in patients without trisomy 21 was 20% (N = 15/77, missing: 6). Atrioventricular septal defect (AVSD) was most frequently seen in 37% (N = 13/35) of the patients with cardiac anomalies. ASD and VSD are among the other cardiac anomalies listed in Table 2.

### 3.3. DO and VACTERL Spectrum

VACTERL spectrum association was seen in 4% (N = 4/112) of the patients with DO. Each of the patients had a combined cardiac anomaly. Due to this small number of patients, VACTERL spectrum was not included in the multivariate logistic regression model. STRICT-VACTERL was seen in one of the four patients and VACTERL-LIKE was seen in the other three. No patients had VACTERL-PLUS association.

### 3.4. Patients Specific Risk Factors for Cardiac Anomalies

Since cardiac anomalies were most the most frequently discovered associated anomaly in children with DO, we examined the patient-specific risk factors for this group. This might improve our ability to identify children preoperatively, and determine which patients require additional diagnostics. Multivariate logistic regression showed that trisomy 21 (OR: 6.5; CI-95% 2.6–16.1) is a significant risk factor for cardiac anomalies in children with DO, as shown in Table 3. Prematurity (OR: 1.6; CI-95% 0.6–4.0) and multiple birth (OR: 1.4; CI-95% 0.1–14.7) were not significant risk factors and were, therefore, excluded from the model. This model has a Nagelkerke R^2^ of 21%.

## 4. Discussion

In our cohort of 112 patients with a DO, 47% (N = 53/112) had another congenital anomaly, with cardiac anomalies being the most common, with an incidence of 31% (N = 35/112), followed by vertebral anomalies, limb anomalies and respiratory tract anomalies. Trisomy 21 was found in 31% (N = 35/112) of the patients, and 57% (N = 20/35) of the patients with a cardiac anomaly were also diagnosed with trisomy 21. Multivariate logistic regression shows that trisomy 21 is a risk factor for cardiac anomalies in patients with DO (OR: 6.5; CI-95% 2.6–16.1). Prematurity and multiple birth are not significant risk factors for cardiac anomalies. In our cohort, the incidence of the VACTERL spectrum was found to be 4% (N = 4/112), with each of the patients having a cardiac anomaly as well.

It is commonly recognized that trisomy 21 and cardiac anomalies are related in the general population, and there is a relationship between these conditions and patients with DO [4,12]. It is still unknown whether cardiac anomalies are associated with DO without the presence of trisomy 21. In our cohort, 31% (N = 35/112) of the patients were diagnosed with trisomy 21, versus 69% (N = 77/112) non-trisomy 21 patients. Within these groups, the incidence of cardiac anomalies was found to be 57% (N = 20/35) in the trisomy 21 group, versus 20% (N = 15/77) in the non-trisomy 21 group. This difference in incidence is similar to previous reports [12,13,14]. Keckler et al., describes an increased risk of cardiac anomalies in patients with trisomy 21 [12]. This is similar to our finding, with trisomy being a significant risk factor for cardiac anomalies in patients with DO, and emphasizes the importance of preoperative screening for cardiac anomalies for patients with trisomy 21. However, with an incidence of cardiac anomalies in 20% of all DO patients in our cohort, a preoperative echocardiography would be justified for all patients with a suspected DO. Another important argument for this routine screening is the mortality rate of about 3% in patients with DO, which can primarily be attributed to the additional anomalies and corresponds to the findings in our own cohort [4,15,16]. However, according to Khan et al., and Short et al., a selective preoperative echocardiography protocol based on chest X-ray, cardiac and respiratory examination might be adequate to screen for cardiac anomalies [5,17].

In addition to cardiac anomalies, our cohort’s vertebral, limb and respiratory tract anomalies each account for about 10% of the patients in the cohort. Atwell et al., find an incidence of vertebral anomalies of 37% in DO patients [9]. This high incidence is in contrast to our incidence of 11% (N = 12/112). However, Gruchalski et al., report an incidence of vertebral anomalies among patients with a duodenal atresia of 2%, which can approach 40% in cases of multiple atresias [18]. Four of the 12 patients in our cohort with vertebral anomalies also have an esophageal atresia or an anorectal malformation. This suggests a potential connection between multiple atresias and vertebral anomalies. However, the sample sizes of these studies do not allow firm conclusions. Regarding the anomalies of the limbs and respiratory tract, Takahashi et al., describe low incidences of limb and respiratory anomalies in a cohort of 31 patients with DO [19].

Trisomy 21 accounts for 86% of all genetic anomalies, as would be expected, making it the most prevalent genetic anomaly. Only a small number of case reports documenting families or individuals with Fanconi anemia, Feingold syndrome, or Cornelia de Lange syndrome have been published [20,21,22]. Therefore, there is no clear association with any of these syndromes in children with DO.

In this cohort, an incidence of 4% (N = 4/112) of VACTERL association is found in patients with DO. This is similar to the incidence of VACTERL reported by Bailey et al., and Choudry et al., in children with DO [23,24]. In the years 2012–2016, the prevalence of VACTERL was 1 in 20,000 live births, which is substantially lower than the 4% incidence found in our cohort [10]. However, the relative rarity of DO and VACTERL limits the ability to investigate the association between DO and VACTERL spectrum due to the small sample sizes. In our cohort, all patients with VACTERL association also have a cardiac anomaly. One explanation for this is that cardiac anomalies are already included in the VACTERL association, making this relation understandable. Due to the fact that these patients with suspected DO would have been eligible for a preoperative echocardiogram, it is already known if there is a cardiac anomaly in these patients. Given the potential association we found, we recommend that patients with DO and a cardiac anomaly without trisomy 21 should receive a VACTERL screening postoperative. A detailed physical examination for dysmorphic features, an ultrasound of the kidneys and sacrum, an X-ray of spine and forearms and additional screening by a clinical geneticist should all be part of this screening [25].

The retrospective design of this study is the main limitation. DO is a relatively rare disorder and it is necessary to include patients from a long time-period to study large numbers of patients. This may cause a bias that influences the results of the study. Another limitation of the retrospective design is the possibility of incomplete cardiac and VACTERL screening among the patients. It is possible that only the most obvious major congenital anomalies were diagnosed and that cases were not screened for additional congenital anomalies that are not immediately evident. Therefore, asymptomatic cardiac or renal congenital anomalies, for example, may have been missed. Moreover, the number of associated anomalies may be underreported as a result of this incomplete screening. An additional prospective multicenter cohort should be considered to conduct a perinatal and perioperative protocol in patients with DO.

## 5. Conclusions

One third of the patients with DO had an associated cardiac anomaly, as well as trisomy 21. Trisomy 21 was found to be a risk factor for cardiac anomalies. VACTERL was found in 4% of the patients, with all of them having a cardiac anomaly. Therefore, we recommend that all patients with DO should receive a preoperative echocardiogram. In patients without trisomy 21 in which a cardiac anomaly is found, a postoperative VACTERL screening should be performed.

## Figures and Tables

**Table 1 children-09-01814-t001:** Patient characteristics.

Variable	Total N = 112 (%)
Male	58 (52%)
Preterm	51 (48%) (missing = 6)
Multiple	5 (5%)
Median age at surgery in days (IQR)	3 (1–6)
Median birthweight in grams (IQR)	2620 (2080–3090) (missing = 7)
Median days hospital stay (IQR)	23 (18–33) (missing = 1)
Median months follow-up (IQR)	33 (6–108) (missing = 1)
Type of obstruction	
Atresia	41 (37%)
Membrane	37 (33%)
Annular pancreas	34 (30%)
Surgery details	
Duodeno-duodenostomy	88 (79%)
Duodeno-jejunostomy	1 (1%)
Duodenoplasty	23 (20%)
Cardiac anomaly	35 (29%)
With trisomy 21	20 (57%)
Without trisomy 21	15 (43%)
Trisomy 21	35 (31%)
Mortality within 30 days	3 (3%)

IQR: Inter quartile range.

**Table 2 children-09-01814-t002:** Overview of associated anomalies.

Type of Anomaly	Total N = 112 (%)	Atresia N = 41	Annular Pancreas N = 34	Web/Membrane N = 37
Number of associated anomalies	53 (47%)	17 (42%)	14 (41%)	22 (60%)
Single	33 (68%)	9 (53%)	8 (57%)	16 (73%)
Multiple	20 (32%)	8 (47%)	6 (43%)	6 (27%)
Cardiovascular	35 (31%)	8 (20%)	11 (32%)	16 (43%)
Atrial septal defect	7 (20%)	1	2	4
Ventricular septal defect	6 (17%)	2	2	2
Atrioventricular septal defect	13 (37%)	1	6	6
Tetralogy of Fallot	3 (9%)	1	1	1
Persistent ductus arteriosus	3 (9%)	1	0	2
Kommerell diverticula	1 (3%)	1	0	0
Isomerism of atrial appendages	2 (6%)	1	0	1
Vertebral	12 (11%)	5 (12%)	3 (9%)	4 (11%)
Scoliosis	4 (33%)	2	1	1
Sacral anomaly	3 (25%)	2	0	1
Anomaly of vertebrae	5 (42%)	3	1	1
Deviating amount of ribs	3 (25%)	0	2	1
Renal	6 (5%)	1 (2%)	4 (12%)	1 (3%)
Pelviureteric junction stenosis	1 (16.6%)	1	0	0
Mono kidney	1 (16.6%)	0	1	0
Horseshoe kidney	1 (16.6%)	0	1	0
Lobulated kidney	1 (16.6%)	0	1	0
Shriveled kidney	1 (16.6%)	0	1	0
Extra renal pyelum	1 (16.6%)	0	0	1
Genito-urinary tract	5 (5%)	1 (2%)	2 (6%)	2 (5%)
Hypospadia	2 (40%)	0	1	1
Uretral obstruction	1 (20%)	0	0	1
Neurogenic bladder in ARM	2 (40%)	1	1	0
Eye, ear and neck	7 (6%)	1 (2%)	1 (3%)	5 (13%)
Hearing loss	6 (86%)	1	1	4
Nystagmus	1 (14%)	0	0	1
Esophageal	3 (3%)	3 (7%)	0 (0%)	0 (0%)
Esophageal atresia with fistula	3 (100%)	3	0	0
Respiratory tract	10 (9%)	3 (7%)	2 (6%)	5 (14%)
Tracheomalacia	3 (30%)	1	0	2
Bronchomalacia	4 (40%)	1	1	2
Tracheobronchomalacia	2 (20%)	0	1	1
Pulmonary hypertension	1 (10%)	1	0	0
Anus	5 (5%)	3 (7%)	1 (3%)	1 (3%)
Anorectal malformation	5 (100%)	3	1	1
Limbs	10 (9%)	4 (10%)	3 (9%)	3 (8%)
Syndactyly	1 (10%)	1	0	0
Clinodactyly	1 (10%)	1	0	0
Anomaly of the thumb	3 (30%)	0	2	1
Hypoplasia forearm	2 (20%)	1	1	0
Clubfoot	2 (20%)	0	0	2
Cleft, lip and palate	1 (1%)	0 (0%)	0 (0%)	1 (3%)
Palatoschisis	1 (100%)	0	0	1
Nervous system	2 (2%)	1 (2%)	0 (0%)	1 (3%)
Microcephaly	2 (100%)	1	0	1
Other	2 (2%)	1 (2%)	0 (0%)	1 (3%)
Polysplenia	2 (100%)	1	0	1
Genetics	42 (38%)	10 (24%)	14 (41%)	19 (50%)
Trisomy 21	35 (85.6%)	9	13	15
Fanconi syndrome	1	0	0	1
Feingold syndrome	1	1	0	0
Cornelia de Lange syndrome	1	0	1	0
22q11 microdeletion	1	0	0	1
Heterotaxie syndrome	1	0	0	1
MCT1 gene defect	1	0	0	1
VACTERL association	4 (4%)	2 (5%)	1 (3%)	1 (3%)
STRICT-VACTERL	1 (25%)	1	0	0
VACTERL-LIKE	3 (75%)	1	1	1
VACTERL-PLUS	0 (0%)	0	0	0

N = Number. ARM = anorectal malformation. VACTERL = vertebral anomalies, anal atresia, cardiac anomalies, tracheo-esophageal fistula, renal anomalies, limb abnormalities [10].

**Table 3 children-09-01814-t003:** Multivariate logistic regression.

Variable	Odds Ratio (OR)	95% Confidence Interval	*p*
Trisomy 21	OR 6.484	[2.611–16.098]	0.000
Prematurity	OR 1.642	[0.666–4.047]	0.281
Multiple birth	OR 1.395	[0.132–14.714]	0.782

## Data Availability

The data presented in this study are available on request from the corresponding author. The data are not publicly available due to privacy.

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
