# Peer review of "The Incidence of Associated Anomalies in Children with Congenital Duodenal Obstruction—A Retrospective Cohort Study of 112 Patients"

_children, 2022, doi:10.3390/children9121814_

Round 1

Reviewer 1 Report

In their retrospective study from two centers in Netherlands the authors investigated the incidence of associated anomalies, a VACTERL-spectrum association and patient-specific risk factors for cardiac anomalies in patients with duodenal obstruction.

Congratulations to the authors on their well-designed and well-written study. I read it with great interest. I have few suggestions for improvement:

1 – Abstract: In methodology section pleas add outcomes of the study.

2. – Key words: The authors should add ‘associated anomalies’ as a key word.

3. – Introduction: The authors mentioned VACTERL spectrum. Please explain what a clinical importance of VACTERL spectrum is and add adequate reference for VACTERL spectrum.

4. – Methodology: Please add date of approval next to Ethic committee reference.

5. – Methodology: Please add primary and secondary outcomes of the study.

6. – Statistical analysis: The authors should mention which statistical test was used to test normality of data distribution.

7. – Results: Row 151/152 – Please revise sentence: ‘in the patients with a web or membrane 62% (N=23/37) was treated by duodenoplasty’ I would suggest: ‘the patients with web or membrane (62%; N=23/37) were treated by duodenoplasty.’

8. – Table 1: Legend:  ’1 Inter quartile range’ – The authors did not mention ‘1’ anywhere in Table. As per style of the journal I would advise ‘IQR’ instead of ‘1’.

9. – Table 3: Please replace ‘p’ instead of ‘p-value’.

Reviewer 2 Report

You consider all the DO as a single malformation, but it is interesting to see if there are different results between atresia, web or annular pancreas, since they have different etiopathogenesis. So, I suggest you analyze the associated anomalies also according to the three different types of duodenal obstruction.

Reviewer 3 Report

The retrospective study by the authors aims to provide a comprehensive overview of the associated anomalies in a cohort of patients with DO. Of the 112 neonates with DO, 47% (N=53/112) had one associated anomaly and 38% (N=20/53) had multiple anomalies. Cardiac anomalies (N=35/112) and trisomy 21 (N=35/112) were present in 31%. In four patients, VACTERL-spectrum was discovered, all with cardiac anomaly. Trisomy 21 was found to be a risk factor for cardiac anomalies (OR:6.5; CI-95%2.6-16.1).

Overall, this is a great study by the authors. However, I have a few suggestions:

Introduction: What did you hypothesize before conducting this research? Please write your hypothesis in 2-3 lines at the end of this section.

Methods: The authors have excluded patients with malrotation as the cause of DO or patients having pyloric atresia. However, it is well-known that malrotation can be an associated anomaly with DO. How did you take care of that? Please mention.

Results: Well written. No changes are needed.

Discussion: Well written. No changes are needed.
